# Evaluation of the Site Form as a Site Productive Indicator in Temperate Uneven-Aged Multispecies Forests in Durango, Mexico

**DOI:** 10.3390/plants11202764

**Published:** 2022-10-19

**Authors:** Jaime Roberto Padilla-Martínez, Carola Paul, José Javier Corral-Rivas, Kai Husmann, Ulises Diéguez-Aranda, Klaus von Gadow

**Affiliations:** 1Department of Forest Economics and Sustainable Land-Use Planning, Faculty of Forest Sciences and Forest Ecology, University of Göttingen, Büsgenweg 1, 37077 Göttingen, Germany; 2Facultad de Ciencias Forestales, Universidad Juárez del Estado de Durango, Río Papaloapan y Boulevard Durango S/N, Durango 34120, Mexico; 3Unidad de Gestión Ambiental y Forestal Sostenible, Departamento de Ingeniería Agroforestal, Escuela Politécnica Superior de Ingeniería, Universidad de Santiago de Compostela, Benigno Ledo s/n, Campus Terra, 27002 Lugo, Spain; 4Faculty of Forestry and Forest Ecology, Georg-August-Universität, Büsgenweg 5, 37077 Göttingen, Germany; 5Department of Forestry and Wood Science, Faculty of AgriSciences, Stellenbosch University, Stellenbosch Private Bag X1, Stellenbosch 7602, South Africa

**Keywords:** site quality, uneven-aged multispecies stands, dominant trees, site productivity index, forest density

## Abstract

Even though the site index is a popular method for describing forest productivity, its use is limited in uneven-aged multispecies forests. Accordingly, the site form (SF) is an alternative measure of productivity to the site index based on the tree height–diameter relationship. Our study aims to evaluate SF as a measure of productivity in the temperate uneven-aged multispecies forests of Durango, Mexico, applying three methods to estimate SF: (i) as the mean height of dominant trees at a reference diameter (SF_H-D_); (ii) as the expected mean height of dominant trees at a reference mean diameter (SF_MH-MD_), and (iii) as the expected height at a reference diameter for a given site (SF_h-dbh_). We assess the effectiveness of the SF based on two hypotheses: (i) the SF correlates to the total volume production, and (ii) the SF is independent of stand density. The SF_H-D_ and the SF_h-dbh_ showed a high correlation with productivity. However, they also did so with density. Contrary to this, the SF_MH-MD_ had a weak correlation with density and productivity. We conclude that the SF is a suitable approach to describe site quality. Nonetheless, its effectiveness as a site quality indicator may be affected according to the method used.

## 1. Introduction

Many decisions aimed at sustainable forest management are based on the assessment of site quality as an indicator of productivity [1,2]. Despite the terms productivity and quality often being used interchangeably to describe the land’s ability to grow trees, they are not synonymous. The former is a quantitative measure of a site’s potential to produce plant biomass. Conversely, the latter is a descriptive measure of site productivity estimated by indirect methods such as the relationship between the height of dominant trees (H; m) and their age [3].

Among the site quality approaches, the site index (SI) is the most popular method used to describe productivity through the H–age relationship. It is therefore frequently used in even-aged stands [4]. The use of the SI is substantiated by the assumptions of a positive relationship between the stand height and volume production [5] and the independence of H from stand density [6]. However, the use of the H–age relationship in uneven-aged multispecies forests has been disputed owing to the age heterogeneity [7] and the lack of distinct growth rings in some tree species, preventing the use of the SI in such instances.

An alternative measure of productivity, based on the relationship between the diameter of dominant trees (D; m) and H, was proposed by Flury in 1929 [8,9] and named site form (SF) by Vanclay and Henry in 1988 [10] to avoid confusion with the SI. In addition to this, the SF has also been called the site productivity index [11]. It has been successfully used as an indicator of productivity in uneven-aged stands [12] and has delivered meaningful results in even-aged forests [13]. Nonetheless, the main feature of SF is its capacity to describe site quality in uneven-aged multispecies forests [14].

The assessment of the usefulness of the SF as a measure of productivity has been carried out through different methods. For instance, the SF as the average H at a specific reference D (SF_H-D_) [10], as the expected mean H (MH) at a reference mean D (MD) (SF_MH-MD_) [12,13], and as the tree height (h) at a reference diameter at breast height (dbh) estimated using a specific h-dbh model for a given site plot (SF_h-dbh_) [14]. Nevertheless, there is no documentation about their comparison, and the results therefore may vary despite being based on the same assumption.

Even though dominant trees are widely used to describe site quality, a universal definition has not emerged to define them. A common approach is to select the dominant trees based on the h or the dbh. Hence, dominance is generally expressed by the 100 tallest or the 100 thickest trees [15] per hectare and selecting a particular definition of dominant trees often relies on how easy it is to measure the required information. However, the number of trees and the method of choosing the dominant trees affect the estimates [16] and thus their effectiveness as a measure of site quality.

Multispecies forests have received increasing scientific and policy attention driven by the hypothesis that forests with a high number of species have an increased productivity and a greater capacity to adapt to climate change impacts [17,18]. Likewise, concerns with regards to the conservation of biodiversity and ensuring the forest’s long-term productivity have incentivized the testing of alternative forest management approaches in uneven-aged stands [19]. However, a high number of tree species complicates selecting one single species as an indicator of productivity [20]. Hence, the SF has been evaluated using the dominant trees, regardless of species, thereby enhancing its practical application [21].

The objective of this study was to evaluate the SF as a measure of productivity for uneven-aged multispecies stands in the temperate forests of Durango, Mexico, using the SF_H-D_, SF_MH-MD_ and SF_h-dbh_ methods. For those methods that required defining the dominant trees, we evaluated them using the 100 tallest and the 100 thickest trees per hectare, regardless of species. The assessment of SF as a measure of productivity was based on two hypotheses: (i) the SF correlates to the total volume production regardless of the stand species composition, and (ii) the SF is independent of stand density.

## 2. Results

### 2.1. H-D, MH-MD and h-dbh Model Fitting

Table 1 gives the estimated parameters and the corresponding goodness-of-fit statistics of the models evaluated. The H-D and MH-MD models explain about 98 percent of the total variance, and their parameters were significant at the five percent level. Therefore, both approaches are suitable for estimating the H or the MH at a specific D or MD. On the other hand, the h-dbh model presented a lower average coefficient of determination (R^2^) and a higher average root mean square error (RMSE). We attributed this result to the high variability of the h and dbh data pairs in each plot, contrary to the H-D and MH-MD models, which only include dominant trees.

### 2.2. Reference Diameter Selection

The H-D and MH-MD models showed a slight reduction of relative error in predictions (RE) at a reference diameter equal to or higher than 40 cm. Similarly, the h-dbh model showed a lower RE value at a dbh of 40 cm (Figure 1) but also presented an increment for trees with a dbh larger than 50 cm, which may be attributed to the lack of trees of this size. We therefore selected a reference diameter of 40 cm for the assessment of the SF for all methods evaluated.

All the predicted curves overlapped well in the paired data sets of H-D, MH-MD and h-dbh and can thus describe the different patterns of H, MH and h growth (Figure 2). Nonetheless, the main drawback of H-D and MH-MD models is that they require measures from permanent plots. Conversely, the h-dbh model may be adjusted using temporal forest inventory plots.

### 2.3. Evaluation of SF as a Site Quality Indicator

The SF_H-D_ and SF_h-dbh_ were positively and significantly correlated with periodic annual increment (PAI; m^3^ ha^−1^ year^−1^), showing Pearson correlation coefficients ranging from 0.2 to 0.7, i.e., from a weak to a strong correlation (Figure 3). In addition, the SF_H-D_ showed a higher correlation using the tallest trees as dominants. Similarly, the use of the Schumacher algebraic difference approach (ADA) model showed a better result than the Schumacher generalized ADA (GADA) model. Hence, the PAI correlates better with the SF using models that predict a steady growth of H or MH and different maximum H or MH values, which seems logical owing to the variation in species mixture. On the other hand, the SF_MH-MD_ showed a low correlation with the PAI. Therefore, estimating forest productivity using the MH or MD in temperate uneven-aged multispecies forests could lead to undesirable results because of the heterogeneity of tree sizes.

Table 2 shows the Pearson correlation between SF and productivity through four levels of species mixture defined by the proportion of conifers. The SF_H-D_ and SF_h-dbh_ correlated with the PAI in stands dominated by broadleaf species, where many of these lack distinguishable growth rings that prevent the use of the SI. Besides this, both methods also showed a good performance in all stand mixture levels evaluated. Contrarily, the SF_MH-MD_ estimated using the 100 tallest trees had a higher correlation in stands dominated by coniferous or broadleaf tree species but not in mixed stands.

Table 3 provides the estimated parameters, with the R^2^ and the RMSE of the linear regression analysis (Figure 3). The SF_MH-MD_ presented the lowest capacity to predict PAI, showing R^2^ values not higher than 0.08. On the other hand, the SF_H-D_ showed a better performance using the 100 tallest trees and the ADA based on the Schumacher model. Furthermore, it presented a homogeneous distribution of the residuals (Figure 4). Similarly, the SF_h-dbh_ showed a high capability to predict PAI, with an R^2^ of 0.49. However, the residuals did not follow a normal distribution (Figure 4).

Figure 5 shows that the SF_MH-MD_ method had the lowest correlation with Reineke’s stand density index (SDI). Contrary to this, SF_H-D_ and SF_h-dbh_ were positively correlated in most of the cases evaluated. Therefore, the estimates of the SF based on individual trees could correlate to the SDI owing to the effect of density on the stem form. Nonetheless, the SF_H-D_ estimated using the 100 thickest trees showed a low correlation. This result may be attributed to the definition of the thickest trees containing a higher number of shade-tolerant species (e.g., trees of *Quercus*, *Arbutus* and *Juniperus)* that can live under closed canopies during the juvenile and adult stages [22].

## 3. Discussion

All evaluated methods showed a different response despite being based on the same assumption of the h-dbh relationship as an indicator of productivity. Therefore, the effectiveness of the SF as a site quality indicator may be affected according to the method, the model, and the specific data set used. Sharma, Amateis and Burkhart [16] reached a similar conclusion by evaluating the performance of seven definitions of H for fitting SI models and concluded that the estimation of the SI using the trees that have always been dominant or codominant over the life of the stand is more precise than the other definitions evaluated. Similarly, the present study shows that the definition of dominant trees may affect the usefulness of SF as a measure of productivity, thus producing unexpected results.

The results show that the 100 tallest trees per hectare exhibit a higher correlation with the PAI than the 100 thickest trees per hectare. Similarly, Vargas-Larreta et al. [23] concluded that the maximum h is a viable estimator of above-ground biomass (Mg ha^−1^) in temperate forests in Durango, Mexico. In addition, Pretzsch, Forrester and Bauhus [24] documented the effect of taller species on the growth of the shorter ones by the competition for soil resources and early acquisition of light. Therefore, the h of the thickest trees may show a low correlation with the PAI and thus might be a non-significant variable [25] or have a negative effect [26] on the prediction of forest productivity.

The high number of tree species in forests complicates the selection of one species or a group of them as an indicator of productivity [20]. Therefore, the assessment of the SF has been carried out by using all trees or the dominant ones to describe site quality as it allows estimating SF for different site conditions [14,21]. In addition, some studies have reported that the relationship between h and dbh varies by forest condition, as it is an adaptive or a passive response to the environment [27,28]. Therefore, the estimation of the SF considering all species may capture the variability of growth patterns of H, MH or h among tree species and thus site conditions. Nonetheless, this approach may only describe whole stand productivity and cannot be used to estimate the growth potential of a specific species.

The ADA based on the Schumacher model had a higher correlation than GADA with the PAI. Likewise, Fu et al. [12] used ADA to test the SF_MH-MD_ in two natural uneven-aged forests of *Larix Olgensis* Henry and *Quercus Mongolica* Fisch in northeastern China, which showed correlation coefficients of 0.55 and 0.79. On the other hand, Molina-Valero et al. [13] reported a correlation of 0.71 in *Pinus radiata* D. Don even-aged stands in the north of Spain using GADA to test the SF_MH-MD_. Therefore, the effectiveness of the SF may vary based on the model used and the type of forest evaluated. Nonetheless, Shen et al. [29] concluded that SF_MH-MD_ was less effective than the SI for estimating site productivity on *Larix olgensis* plantations. Hence, the SI may be a better measure of site productivity in planted forests where the stand age is known [11].

The volume production in multispecies forests is a response that depends on site conditions, structural attributes, species composition, forest density and the interaction between these factors [30,31,32]. Hence, it could be expected that the regression analysis does not show a R^2^ higher than 0.49. Similarly, an alternative index to the SI for irregular stands, developed by Berrill and O’Hara [33], expressed 32 percent of the variance of stand volume increment. Nonetheless, this index showed a better performance than the SI and a significant effect on the increment of individual trees. Likewise, the SF_h-dbh_ and the SF_H-D_, estimated by the Schumacher GADA model and the 100 tallest trees, may be used in combination with other variables to describe the forest productivity in stands where the SI cannot be estimated or is not a significant variable to predict the stand volume increment.

Our study shows that most of the SF evaluated correlated positively with the SDI, especially for the SF_H-D_ and the SF_h-dbh_. However, some studies have found that the estimation of the SI is affected by stand density, thus questioning the assumption of independence [34,35,36]. Asthon and Kelty [15] also documented that very high or low density may influence the h growth of species, conditions that were sampled in the current study that may therefore increase the correlation between the SF and the SDI. In addition, the h-dbh ratio (calculated by dividing the h by the dbh) has been described as a measure related to the stand density that increases in closed spaces [37,38,39], affecting dominant and codominant trees [40,41]. Therefore, stand density may influence the estimation of the SF. In addition, other studies have found that SF correlates positively to stand basal area [10,14,20], a measure commonly used to describe stand density.

On the other hand, alternative approaches describe forest yield based on stand density. For instance, the hypotheses of Wiedermann [42], Assmann [43] and Mar:Moller [44] assume a correlation between volume growth and stand basal area. These hypotheses suggest that timber production is constant until the stand reaches its maximum density, a pattern evaluated in several studies carried out in Mexican temperate forests [25,45,46]. Therefore, the correlation between the SDI and the SF seems logical, as both correlate with timber production. In addition, stand density has also been used as a measure of forest productivity in uneven-aged stands as an alternative to the SI [33].

Even though there are several SI models developed for Mexican forests [47,48,49], their evaluations focus on the predictions of H and not on their capability to describe stand productivity. Therefore, their application may lead to low accuracy on stand yield predictions, likely due to the mixture of species and age classes that complicate the application of the SI [4]. On the other hand, the SF_h-dbh_ and the SF_H-D_, estimated by the Schumacher GADA model and the 100 tallest trees, were positively correlated to the PAI on different stand mixtures. In addition to this, the SF requires h and dbh measurements, which are variables available from routine inventories and compatible with existing forest inventory data [12]. Therefore, this highlights its applicability as an indicator of quality in uneven-aged multispecies forests in Durango, Mexico.

## 4. Materials and Methods

The dataset used in this study derives from a network of 423 permanent plots established by the Universidad Juárez del Estado de Durango [50]. These plots cover a great variety of stands, with different levels of density and productivity in Durango’s temperate forests, where the predominant stand condition is uneven-aged pine-oak communities [51]. The location of the permanent plots is given in Figure 6.

The 423 plots were established in 2007 and re-measured in 2012. Of these, a total of 107 were measured a third time in 2017. Each plot covers an area of 2500 m^2^ (50 m × 50 m). Within these plots, all trees with a dbh equal to or larger than 7.5 cm were measured. The dbh was measured by caliper and the h by digital hypsometer (Vertex IV). The scientific name, the height to the living crown (m), and the azimuth (degrees) and distance (m) from the center of the plot were also recorded.

A total of 73 tree species were sampled: 34 *Quercus*, 19 other broadleaf species, 15 *Pinus* and five other coniferous species. *Pinus* was the most prominent genus according to the importance value index (the sum of the relative values of the number of individuals, basal area and frequency per species), making up approximately 50 percent of the total value. Additionally, the plots showed a right-skewed distribution for the dbh and the h, most of which were made up by small-size individuals. Table 4 shows a summary of the permanent plots used in this study.

### 4.1. H-D, MH-MD and h-dbh Models

To select a suitable base model for evaluating the SF, we fitted several models commonly used in studies of the SI (Bertalanffy-Richards, Hossfeld, Korf, Schumacher). Among them, the Schumacher model [52] showed the best fit for our data. Therefore, we selected the Schumacher function as a base model to develop the H-D, MH-MD and h-dbh models. This model has been used in various studies carried out in Mexican temperate forests that evaluated site productivity or individual tree growth [45,53,54]. The mathematical formulation of the Schumacher model is given below:(1)$$h=1.3{\;+\;e}^{{(\beta }_{0}{+\beta }_{1}{*\mathrm{dbh}}^{-1})}$$
where β_0_ and β_1_ are the regression parameters.

We used Equation (1) to build the specific nonlinear h-dbh models for each plot as a basis for estimating the SF_h-dbh_. These specific h-dbh models were fitted using the *nls* function of R [55]. On the other hand, we adjusted the H-D and MH-MD models using the ADA [56] and the GADA [57] based on the Schumacher model. These approaches describe the development of H or MH by a set of H-D or MH-MD curves, allowing for the classification of the stand according to the Eichhorn rule. The mathematical formulations of ADA and GADA based on the Schumacher model are given below:

Schumacher ADA,
(2)$$H_{2}{\;\mathrm{or}\;\mathrm{MH}}_{2}=1.3\;+\;((H_{1}\;\mathrm{or}\;MH_{1})-1.3)\left[\frac{\exp\left({-\beta }_{1}{/(D}_{2}\;\mathrm{or}\;MD_{2})\right)}{\exp\left({-\beta }_{1}/(D_{1}\;\mathrm{or}\;MD_{1})\right)}\right]$$
and Schumacher GADA [58],
(3)$$H_{2}{\;\mathrm{or}\;\mathrm{MH}}_{2}=1.3{\;+\;\exp(X}_{0}\;+\;\left(\beta_{0}{\;+\;\beta }_{1}{\;*\;X}_{0}\right){{(D}_{2}\;\mathrm{or}\;MD_{2})}^{-1})$$
where $X_{0}=\frac{\left(\ln\left((H_{1}\;\mathrm{or}\;MH_{1})\;-\;1.3\right){-\beta }_{0}{\left(D_{1}\;\mathrm{or}\;MD_{1}\right)}^{-1}\right)}{{1+\beta }_{1}{\left(D_{1}\;\mathrm{or}\;MD_{1}\right)}^{-1}}$, $H_{n}$, $MH_{n}$, $D_{n}$ and $MD_{n}$ are the H, MH, D, and MD at time *n,* and β_0_ and β_1_ are the regression parameters.

We fitted Equations (2) and (3) using the 100 thickest and the 100 tallest trees per hectare as the definitions of dominant trees. Both definitions included only re-measured live trees during the whole study period and did not distinguish among species. Mohamed et al. [21] used a similar definition of dominant trees to develop H-D curves using the tallest trees, regardless of species, allowing for the comparison of stands with different species compositions.

We used a nonlinear mixed effects modeling approach for the development of the H-D and MH-MD models, simultaneously considering global parameters for the whole population (fixed-effects) and specific parameters for each tree or plot (random effects). As well as this, the autocorrelation was corrected using an autoregressive structure (AR1). We used the function *nlme* of the NLME package [59] of R [55] to fit the H-D and MH-MD models.

Two goodness-of-fit statistics of H-D and MH-MD models were estimated: the adjusted coefficient of determination,
(4)$$R^{2}=1-\left\lceil \frac{n-1\sum_{i=1}^{n}{\left(y_{i}-{\hat{y}}_{i}\right)}^{2}}{n-p\sum_{i=1}^{n}{\left(y_{i}-\bar{y}\right)}^{2}}\right\rceil ,$$
and the root mean square error,
(5)$$\mathrm{RMSE}=\sqrt{\frac{\sum_{i=1}^{n}{\left(y_{i}-{\hat{y}}_{i}\right)}^{2}}{n-p}},$$
where $y_{i}$, ${\hat{y}}_{i}$, and $\bar{y}$ are the observed, predicted, and mean values of the dependent variable, respectively; n is the total number of observations, and p is the number of parameters used to fit the models. R^2^ indicates the proportion of the variance of the dependent variable explained by the model, while RMSE indicates the precision of the estimates. We estimated the statistics of the h-dbh model as the average of the R^2^ and the RMSE of the specific h-dbh models.

### 4.2. Reference Diameter Selection

Do et al. [14] suggested that the reference diameter should be a diameter commonly found in the forest, not too small so as to increase the difference among stands and provide higher accuracy to predict H, MH or h. Therefore, we defined the reference D, MD and dbh through a graphical analysis of the evolution of the RE (Equation (6)) at different diameters, an approach that allows for the identification of the reference diameters with higher accuracy predictions.
(6)$$\mathrm{RE}=\frac{\mathrm{RMSE}}{\bar{y}}*100,$$
where all variables are as previously defined.

In addition to this, we carried out a graphical analysis of the expected H and MH at a specific reference diameter overlaid on the H-D and MH-MD data pairs to evaluate the performance of the models. The curves derived from the h-dbh model were developed based on the methodology proposed by Do et al. [14], which requires grouping the plots into SF_h-dbh_ classes and building a h-dbh model for each SF_h-dbh_ class.

### 4.3. Evaluation of SF as a Site Quality Indicator

A total of nine SF measures were estimated for each plot and inventory period: four SF_H-D_, four SF_MH-MD_ and one SF_h-dbh_. The SF_H-D_ and the SF_MH-MD_ were estimated four times as we evaluated two definitions of dominant trees and two models.

Timber volume was calculated using specific volume equations for each species and region [60]. These estimate the volume of stems and branches, thus adding up to the total tree volume:(7)$$\mathrm{Tree}’{s\;\mathrm{volume}=a}_{0}{\mathrm{dbh}}^{a_{1}}h^{a_{2}}{+a}_{3}{\mathrm{dbh}}^{2},$$
where $a_{0}$, $a_{1}$, $a_{2}$ and $a_{3}$ are specific parameters by species and region [60]. Subsequently, the individual tree volume increment was estimated by taking the differences in volume between measurements and dividing this by the length of the period (five years). Lastly, it was extrapolated to one hectare and summed at the plot level, resulting in the PAI.

We defined Reineke’s index [61],
(8)$$\mathrm{SDI}=N{\left(\frac{\mathrm{Dq}}{25.4}\right)}^{1.605},$$
as a measure of density to assess the relationship between the SF and stand density. This index assumes a maximum number of trees that may coexist in a stand according to the number of trees per hectare (N) and the quadratic mean diameter (Dq, cm), thus describing stand density as a function of the maximum density achievable and these two variables.

We evaluated the SF-productivity relationship using Pearson’s correlation test between the SF and the PAI. This evaluation was also conducted for four stand mixture levels, defined by the relative proportion of coniferous species’ individuals. Additionally, we evaluated the performance of the SF as a measure of productivity through linear regression analysis. Lastly, we tested the independence of the SF from stand density using Pearson’s correlation test.

## 5. Conclusions

The SF may be used to estimate the productive potential of stands with different tree species and size classes. It is therefore not possible to reject the hypothesis of the correlation between SF and volume production. This study presents evidence that the SF_h-dbh_ and the SF_H-D_, estimated by the Schumacher GADA model and the 100 tallest trees per hectare, are suitable approaches to describe the site productivity potential of temperate uneven-aged multispecies forests in Durango, Mexico. Nonetheless, the SF correlated positively with the SDI in most of the cases evaluated, likely due to the effect of density on the h-dbh ratio and thus rejecting the hypothesis of the independence of SF from stand density.

## Figures and Tables

**Figure 1 plants-11-02764-f001:**
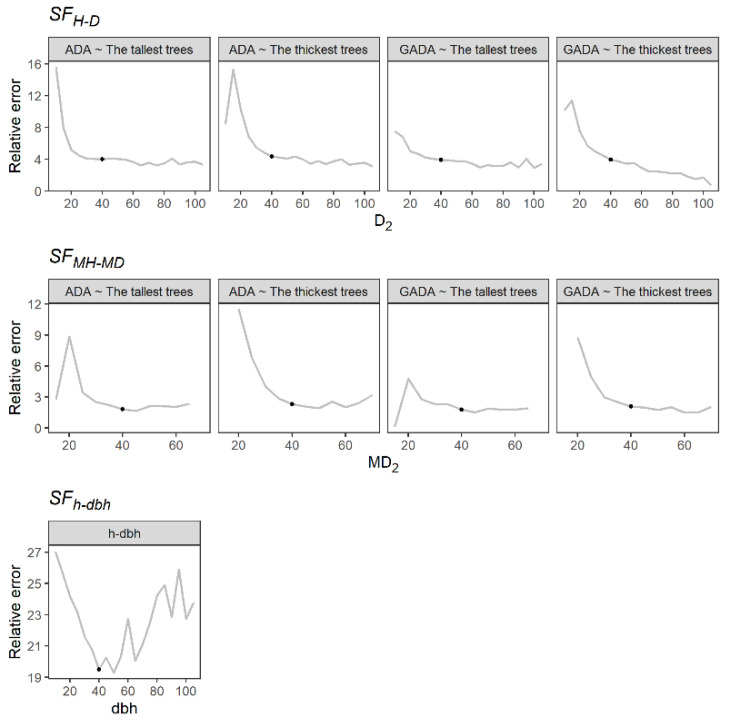
Relative error in height predictions at different reference diameters. The black point indicates the selected reference diameter.

**Figure 2 plants-11-02764-f002:**
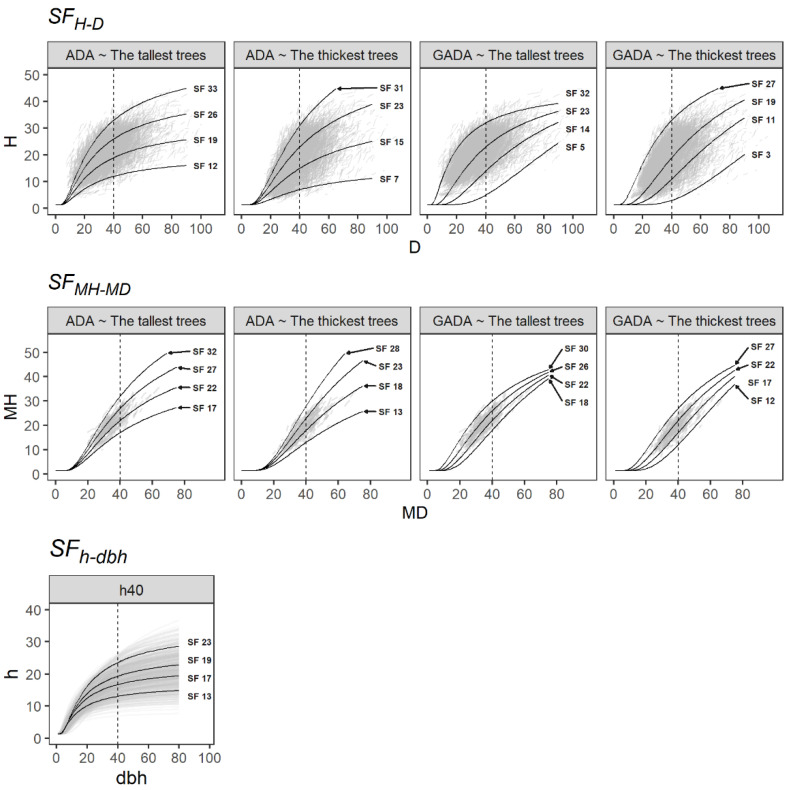
Site form curves based on the H-D, MH-MD and h-dbh models derived from the SF methods evaluated. The dash line indicates the reference diameter.

**Figure 3 plants-11-02764-f003:**
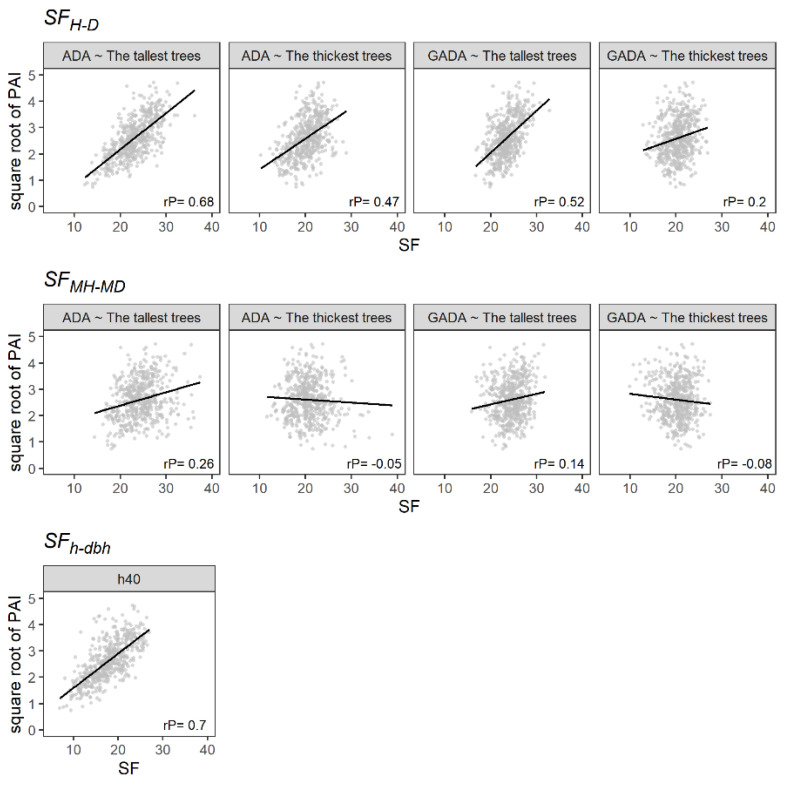
The relationship between the SF and the square root of the PAI (m^3^ ha^−1^ year^−1^). The black line illustrates the linear regression, and rP is the Pearson correlation.

**Figure 4 plants-11-02764-f004:**
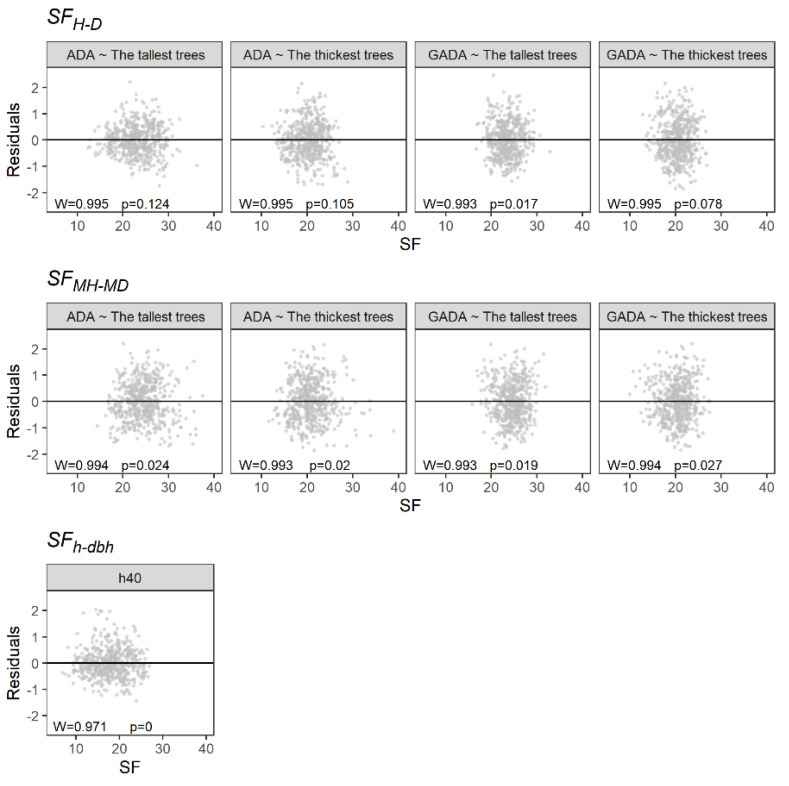
Residual plot of the linear regression analyses. W is the Shapiro–Wilk test, and *p* is the *p* value.

**Figure 5 plants-11-02764-f005:**
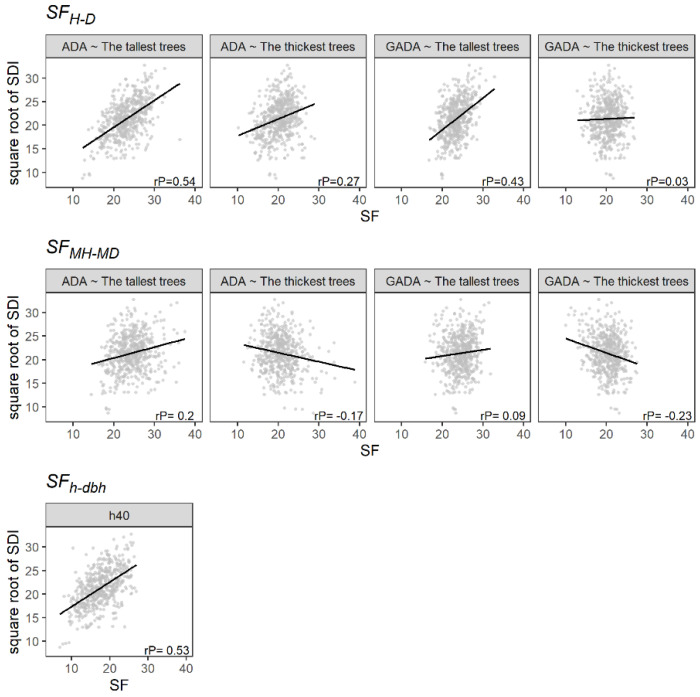
The relationship between the SF and the square root of the Reineke’s stand density index (SDI). The black line illustrates the linear regression, and rP is the Pearson correlation.

**Figure 6 plants-11-02764-f006:**
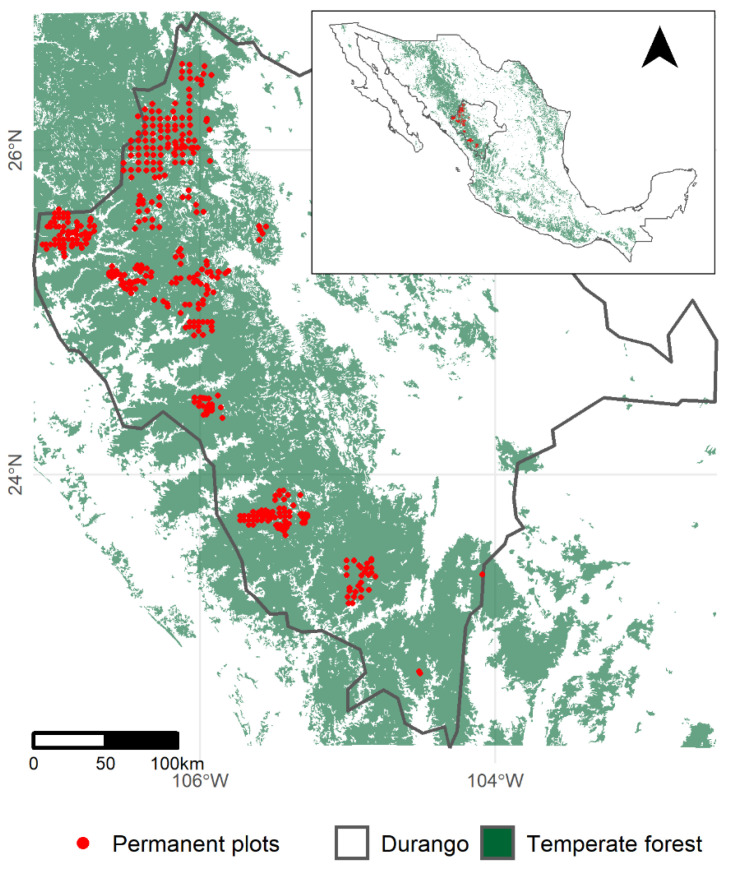
Location of the 423 permanent plots.

**Table 1 plants-11-02764-t001:** Parameter estimates and goodness-of-fit statistics.

Method	Model	Trees Selected	Parameter	Estimator	SE	R^2^	RMSE
SF_H-D_	Sc. ADA	The 100 tallest trees	$\beta_{1}$	23.02	0.23	0.98	0.86
SF_H-D_	Sc. GADA	The 100 tallest trees	$\beta_{0}$	534.60	38.45	0.98	0.83
			$\beta_{1}$	−144.44	9.83		
SF_H-D_	Sc. ADA	The 100 thickest trees	$\beta_{1}$	39.67	0.36	0.98	0.89
SF_H-D_	Sc. GADA	The 100 thickest trees	$\beta_{0}$	845.25	112.33	0.99	0.77
			$\beta_{1}$	−209.41	26.29		
SF_MH-MD_	Sc. ADA	The 100 tallest trees	$\beta_{1}$	43.34	0.99	0.99	0.49
SF_MH-MD_	Sc. GADA	The 100 tallest trees	$\beta_{0}$	347.70	66.13	0.99	0.43
			$\beta_{1}$	−91.35	15.29		
SF_MH-MD_	Sc. ADA	The 100 thickest trees	$\beta_{1}$	63.17	1.30	0.99	0.52
SF_MH-MD_	Sc. GADA	The 100 thickest trees	$\beta_{0}$	394.75	75.84	0.99	0.44
			$\beta_{1}$	−100.81	16.54		
SF_h-dbh_	Schumacher	All trees				0.62	2.86

R^2^ is the adjusted coefficient of determination, RMSE is the root mean square error and Sc. ADA and Sc. GADA are the algebraic difference approach and its generalization based on the Schumacher model, respectively.

**Table 2 plants-11-02764-t002:** Pearson correlation between each SF evaluated and the square root of the PAI by stand mixture levels.

			Pearson Correlation			
Method	Model	Trees Selected	Coniferous Individuals’ Proportion			
			0–0.25	0.26–0.5	0.51–0.75	0.76–1
SF_H-D_	Sc. ADA	The 100 tallest trees	0.74	0.71	0.58	0.75
SF_H-D_	Sc. GADA	The 100 tallest trees	0.42	0.46	0.44	0.6
SF_H-D_	Sc. ADA	The 100 thickest trees	0.21	0.53	0.43	0.48
SF_H-D_	Sc. GADA	The 100 thickest trees	0.05	0.4	0.33	0.4
SF_MH-MD_	Sc. ADA	The 100 tallest trees	0.37	0.15	0.14	0.35
SF_MH-MD_	Sc. GADA	The 100 tallest trees	0.28	−0.01	0.08	0.21
SF_MH-MD_	Sc. ADA	The 100 thickest trees	−0.08	−0.16	−0.08	−0.04
SF_MH-MD_	Sc. GADA	The 100 thickest trees	−0.05	−0.15	−0.11	−0.08
SF_h-dbh_	Schumacher	All trees	0.41	0.79	0.7	0.66

Sc. ADA and Sc. GADA are the algebraic difference approach and its generalization based on the Schumacher model, respectively.

**Table 3 plants-11-02764-t003:** Linear regression analysis per each SF evaluated.

Method	Model	Trees Selected	Intercept	Slope	R^2^	RMSE
SF_H-D_	Sc. ADA	The 100 tallest trees	−0.597	0.139	0.46	0.552
SF_H-D_	Sc. GADA	The 100 tallest trees	−1.146	0.160	0.27	0.642
SF_H-D_	Sc. ADA	The 100 thickest trees	0.242 ^ns^	0.117	0.22	0.665
SF_H-D_	Sc. GADA	The 100 thickest trees	1.350	0.062	0.04	0.738
SF_MH-MD_	Sc. ADA	The 100 tallest trees	1.391	0.050	0.07	0.729
SF_MH-MD_	Sc. GADA	The 100 tallest trees	1.621	0.040	0.02	0.746
SF_MH-MD_	Sc. ADA	The 100 thickest trees	2.847	−0.012 ^ns^	0.00	0.753
SF_MH-MD_	Sc. GADA	The 100 thickest trees	3.053	−0.022 ^ns^	0.01	0.751
SF_h-dbh_	Schumacher	All trees	0.290	0.131	0.49	0.541

R^2^ is the adjusted coefficient of determination, RMSE is the root mean square error, Sc. ADA and Sc. GADA are the algebraic difference approach and its generalization based on the Schumacher model, respectively. ns is a non-significant parameter (α = 0.05).

**Table 4 plants-11-02764-t004:** Summary statistics of the 423 plots used in this study.

Variable	1st Inventory (423 Plots) 2007		2nd Inventory (423 Plots) 2012		3rd Inventory (107 Plots) 2017	
	Mean (Min–Max)	S.D.	Mean (Min–Max)	S.D.	Mean (Min–Max)	S.D.
S	8 (2–15)	2	8 (2–16)	3	8 (2–14)	2
N	622 (120–2148)	272	606 (108–2060)	269	692 (140–2092)	318
dbh	17.9 (7.5–104)	10.6	18.8 (7.5–105.5)	10.8	18.7 (7.5–106)	10.9
h	10.7 (1.5–48.2)	5.6	11.8 (1.5–49.4)	5.8	12.9 (2.1–49.7)	6
G	21.4 (3.1–53.9)	8.09	22.3 (3.7–55.9)	8.51	25.2 (4.4–58.7)	10.5
V	194.7 (12.1–697.3)	109.0	215.5 (15.7–786.2)	119.6	261.6 (19.7–851.3)	151.1
Dq	21.6 (12.4–51.0)	4.7	22.3 (12.9–52.2)	4.8	22.1 (13.4–41.8)	4.90
SDI	449.5 (71.7–1047.5)	155.4	462.8 (84.4–988.6)	162.2	523.2 (95.9–1044.3)	199.7
PAI			7.3 (0.7–23.5)	4.2	9.2 (0.6–22.7)	4.9

S is the number of tree species in the plot; N is the number of trees per hectare; dbh is the diameter at breast height (cm); h is the total tree height (m); G is the stand basal area (m^2^ ha^−1^); V is the estimated aboveground stand volume (m^3^, ha^−1^); Dq is the quadratic mean diameter (cm); PAI is the periodic annual increment (m^3^ ha^−1^ year^−1^); SDI is the Reineke’s stand density index; and min, max and S.D. are the minimum, maximum and standard deviation values of each variable.

## Data Availability

Not applicable.

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
