# Peer review of "Evaluation of the Site Form as a Site Productive Indicator in Temperate Uneven-Aged Multispecies Forests in Durango, Mexico"

_plants, 2022, doi:10.3390/plants11202764_

Round 1

Reviewer 1 Report

The authors try to evaluate the site productivity of uneven aged multispecies forests through site form index based. The topic is interesting and the results are used for assessing the site productivity of uneven aged forests because of unknown age of uneven aged forest if we used the common method site index. After reading the paper, I found the paper is not clear, especially for the method section.

 1. Are the data divided into fit and validation data sets? When we build a model, I think we should validate the model using the other data.

 2. Firstly, the authors should define the reference diameter before building the SF model (eq. 2 and 3). But the reference diameter is defined based on the RMSE, which was calculated from the model (eq. 2 or 3). So how did the authors estimate the models? I think the authors need tell the readers more detailed.

 3. Why regressing SF against volume increment? The authors should clarify.

 Some specific comments:

 Table 3: As we can find the highest R2 was only 0.46, it means that 54% variance of site productivity cannot be reflected in the model. So how can we use the model? I do no think the model can be used for assessing the site productivity. Are the models Sc. ADA and Sc. GADA are suitable for regressing SF against volume increment?

Line 276: build the d-dbh model by each plot? Why not building the d-dbh model using the whole data? What is each d-dbh model used for?

 Line 288-289: remeasured live trees during the three inventories?

Author Response

Dear reviewer.

Beforehand, we appreciate your constructive feedback. A point-by-point explanation of how we addressed each of your comments is provided in the attached document.

Reviewer 2 Report

The manuscript has an unusual structure when the chapter “Materials and Methods” does not follow the “Introduction”.

Pg.2, Ln 69, Pg.7, Ln 183, 184, Pg.12, Ln 347 – it should be right „100 thickest (tallest) trees per hectare“.

Pg.5, Table 2 – the values in the Table header must be modified 0-0.25; 0.26-0.5; 0.51-0.75; and 0.76-1.00

Pg.6, Table 3 – according to what level of significance were the mentioned differences insignificant?

Pg.9 to 12, Ln. 240 to 343 – The chapter „Materials and Methods“ move behind „Introduction“.

Pg.12, Conclusions – It should be added whether the 2 hypotheses were confirmed or not.

Author Response

(The authors gave the same response as above.)

Round 2

Reviewer 1 Report

The authors have addressed all my concerns.